# Biodiversity and Phylogeny of North Atlantic Euphrosinidae (Annelida)

Rowan A. Batts [1,†], Karsyn N. Whitman [1], Karin Meißner [2] and Kevin M. Kocot [1,3,*]

1  Department of Biological Sciences, University of Alabama, Tuscaloosa, AL 35487, USA
2  Senckenberg am Meer, Deutsches Zentrum für Marine Biodiversitätsforschung, Martin-Luther-King Platz 3, 20146 Hamburg, Germany
3  Alabama Museum of Natural History, University of Alabama, Tuscaloosa, AL 35487, USA
*  Correspondence: kmkocot@ua.edu; Tel.: +1-(205)-348-4052
†  Present address: Department of Marine Sciences, University of Connecticut, Groton, CT 06340, USA.

**Abstract:** Euphrosinidae (Amphinomida) is a clade of generally small, short but stout annelids characterized by long, calcareous chaetae that may be distally forked or ringent. Little is known about the diversity of Euphrosinidae from the North Atlantic and the phylogeny of the group has received little attention. Here, we examined 59 specimens of Euphrosinidae (primarily from the IceAGE I and II cruises) and sequenced fragments of the mitochondrial 16S rDNA and nuclear 28S rDNA genes to improve understanding of euphrosinid diversity in the North Atlantic and gain insights into euphrosinid phylogeny. Maximum likelihood analysis of 28S + 16S recovered *Euphrosine* as a 'basal' paraphyletic grade; a clade containing *E. armadillo* (plus other unidentified specimens) was sister to *Euphrosinopsis* + *Euphrosinella* while a clade containing *E. aurantiaca* and *E. foliosa* (plus three unidentified species) was recovered sister to all other sampled Euphrosinidae species. Species delimitation analyses based on 16S sequences identified between 14 and 11 species of Euphrosinidae with as many as ten distinct species from the North Atlantic. The IceAGE material investigated includes one new species of *Euphrosinopsis* and at least one new species of *Euphrosinella*. Unfortunately, because most of this material was preserved in ethanol, we were unable to characterize key features needed for adequate species descriptions. Additionally, PCR contaminants from presumed gut contents suggest that some euphrosinids eat other annelids, namely Cirratulidae and Syllidae.

**Keywords:** DNA barcoding; *Euphrosine*; *Euphrosinopsis*; *Euphrosinella*; 16S; 28S

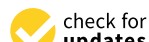



## 1. Introduction

While molecular studies of annelid diversity and phylogeny have increased, many groups including Euphrosinidae (Amphinomida) remain understudied. Euphrosinidae Williams 1852 is a particularly understudied group with relatively little work focusing on its diversity and evolution. Euphrosinids are wide compared to their length, dorso-ventrally flattened, and orange, peach, or tan-colored in life. They have numerous long, calcareous chaetae [1] that are often distally forked or ringent (simple, subdistally expanded chaetae with a narrow slit that are internally crenulated or serrated; Figure 1). These chaetae give them the appearance of "fuzzballs" and often obstruct study of underlying anatomical structures such as branchiae. Euphrosinids have been collected from as deep as 3570 m but other species can be relatively easily collected from shallow waters by SCUBA diving [2]. In terms of body size, *Euphrosinopsis antipoda* is one of the smallest species at 0.9–3.1 mm long and 0.8–1.2 mm wide (excluding chaetae) while *Euphrosine monroi*, one of the largest euphrosinid species, can reach 19 mm long and 4.5 mm wide excluding chaetae [2].

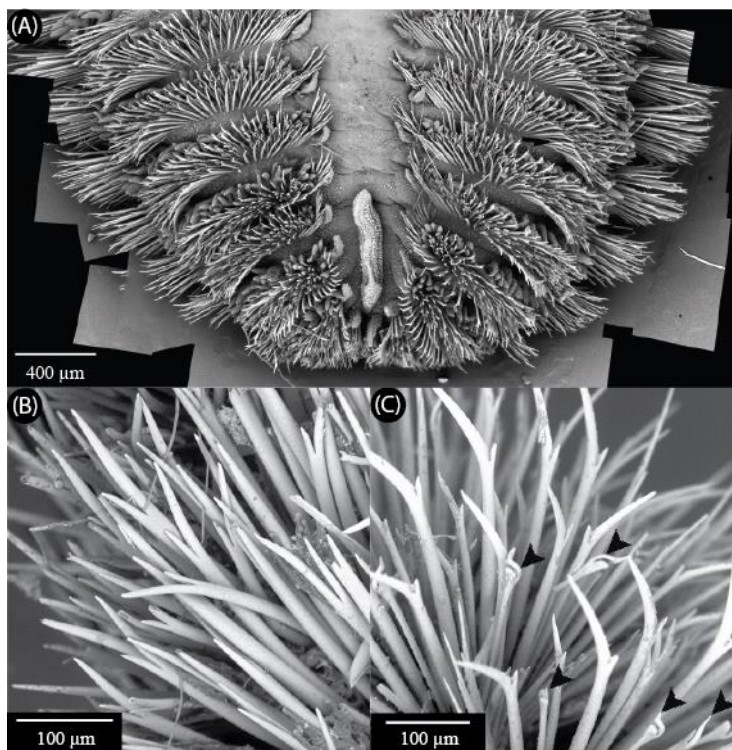

**Figure 1.** SEM images showing chaetae of Euphrosinidae species. (**A**) *Euphrosine aurantiaca* anterior end showing organization of chaetae on the body and caruncle. Image is a composite of several SEM photos generated in Microsoft Image Composite Editor. (**B**) *Euphrosine* sp. indet. (DZMB-HH 43974-A) with distally forked chaetae. (**C**) *Euphrosinopsis* sp. indet. (DZMB-HH 50004-K) with distally forked and ringent chaetae. Ringent chaetae are indicated by arrowheads.

Euphrosinidae includes approximately 65 named species and four genera [3] of which only three were sampled in this study: *Euphrosine* Lamarck 1818 is the largest genus with 55 globally distributed species, *Euphrosinella* Detinova 1985 includes only two species (one from California and one from Antarctica), *Euphrosinopsis* Kudenov 1993 includes three Antarctic species (one of which has also been documented in the Sea of Okhotsk), and *Palmyreuphrosyne* Fauvel, 1913 includes two species (one from Three Kings Island, New Zealand and one from the Azores) [4,5].

*Euphrosinopsis* is the most easily identified genus as it has one pair of eyes where *Euphrosinella* and *Euphrosine* have two pairs [2]. *Euphrosine* and *Euphrosinella* species are often morphologically similar with the number of prostomial appendages (three in *Euphrosine* and five in *Euphrosinella*) being the most reliable distinguishing characteristic. Notably, both of the described species of *Euphrosinella* were originally classified as *Euphrosine* [2,5]. Branchiae and chaetae are important taxonomic characters at the species level but are not very useful when placing a specimen into a particular genus. The genus *Palmyreuphrosyne*, which was not sampled in this study, is distinguished by more flattened notochaetae and smooth paleae compared to the other three genera [6]. Differences between the three sampled genera are summarized in Table 1.

Identifying euphrosinids based on morphological characteristics is challenging due to their small size and their long chaetae, which can make examining their branchiae, a key diagnostic feature at the species level, difficult. Prostomial appendages, which are also key diagnostic features, can become damaged during sampling or handling, further complicating euphrosinid identification. Preservation in ethanol or RNAlater for use in molecular studies often causes key features of soft anatomy such as prostomial appendages and branchiae, to shrivel up and become brittle, breaking easily. These challenges make

DNA barcoding a valuable tool for the study of euphrosinid diversity. DNA barcoding can also be useful to distinguish between cryptic species [7,8].

**Table 1.** Distinguishing features between *Euphrosine*, *Euphrosinopsis*, and *Euphrosinella*.

| Genus | Eyes | Prostomium | Caruncle | Chaetae |
|---|---|---|---|---|
| *Euphrosine* | Two pairs of eyes | Prostomium with three appendages | Caruncle attached to body wall | Simple bifurcate chaetae; ringent chaetae and aciculae may be present |
| *Euphrosinopsis* | One large pair of eyes | Prostomium with five appendages | Caruncle not attached to the body wall for most of its length | Simple bifurcate chaetae; ringent chaetae may be present |
| *Euphrosinella* | Two pairs of eyes | Prostomium with five appendages | Caruncle not attached to the body wall for most of its length | Simple bifurcate chaetae |

Most studies on euphrosinids to date have been primarily focused on taxonomy [2,3,9–11] and just a few studies have generated DNA sequence data for the clade. One study on the diversity of polychaetes in Antarctica generated 16S DNA barcodes for several specimens of *Euphrosinella* cf. *cirratoformis* and *Euphrosinopsis* cf. *antarctica* [8]. Another study examining relationships between families of Amphinomida sequenced several markers including 28S from *Euphrosine armadillo* and *Euphrosine foliosa* [12]. Prior to the present work, NCBI GenBank had just 41 entries for Euphrosinidae, representing 5 species and several entries identified only to the genus level. The availability of existing 16S and 28S sequences for Euphrosinidae and the phylogenetic information content of these molecular markers makes them good barcoding genes for the study of euphrosinid diversity and phylogeny. Due to its relatively rapid rate of evolution, 16S is generally a good molecular marker for distinguishing between annelid species; on the other hand, 28S is better for resolving higher-level evolutionary relationships because it is more conserved [13].

Here, we assessed North Atlantic Euphrosinidae biodiversity and euphrosinid phylogeny by analyzing sequences from 82 euphrosinids. Taxon sampling was focused on specimens from the IceAGE (Icelandic Animals: Genetics and Ecology) I and II cruises [14], which accounts for 59 of the specimens sampled. Our phylogenetic and species delimitation analyses provide insight on euphrosinid phylogeny (chiefly the non-monophyly of the genus *Euphrosine*), expand the known range of some taxa, and present evidence for multiple new species [15].

## 2. Materials and Methods

### 2.1. Sample Collection

Specimens were primarily collected from the North Atlantic during the 2011 and 2013 IceAGE (Icelandic Animals: Genetics and Ecology) I and II cruises using an epibenthic sled or box corer [14] (Figure 2, Supplementary Material Table S1). Lots of euphrosinids sorted from these samples were given DZMB numbers, which are field collection numbers rather than museum catalog numbers, and individual specimens were distinguished by adding a -A, -B, -C, and so on suffix. Additional specimens were collected from the Amundsen Sea and Ross Sea (Antarctica) during the NBP 12-10 cruise and *Euphrosine aurantiaca* was collected as part of the Diversity Initiative for the Southern California Ocean (DISCO) project. Specimens preserved in 96% ethanol were handled following a "cooling chain" to help ensure the preservation of DNA [16]. Other specimens were fixed in 4% formaldehyde and then transferred to 70% ethanol to be used in later morphological analysis.

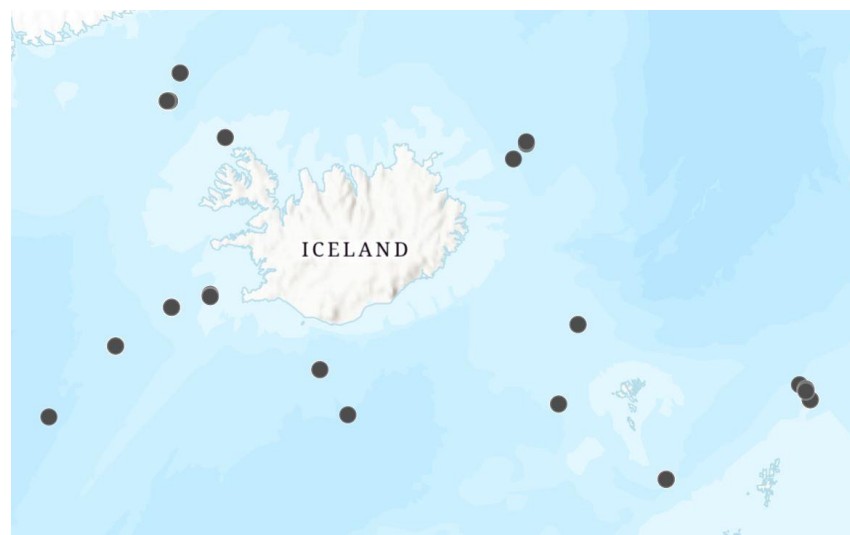

**Figure 2.** Sampling locations for euphrosinids collected during the IceAGE I cruise.

## 2.2. Imaging

Specimens were imaged using an Olympus SZX 16 stereomicroscope with an Olympus SC50 camera and Olympus CellSens imaging software. To study and image chaetae, select specimens were also imaged using a Phenom Pro Scanning Electron Microscope (SEM). Because this is a low accelerating voltage, low-vacuum SEM, most specimens were simply taken out of 96% ethanol and allowed to dry in a Petri dish prior to being mounted on an SEM stub and imaged without critical point drying or sputter coating.

## 2.3. DNA Extraction, Amplification, and Sequencing

DNA was extracted using the Omega Bio-tek EZNA MicroElute Genomic DNA kit using the procedures described by the manufacturer. Polymerase Chain Reaction (PCR) was performed using a 25 μL volume reaction containing 9.5 μL ddH2O, 12.5 μL AMRESCO 2X Hot Start Taq Master Mix, 1 μL of each primer (10 μM), and 1 μL template DNA. Reactions yielding very faint PCR bands were re-done with 2 μL template DNA. The 16S arL/brH primers [17] were used on every sample to amplify a ~540 bp fragment of the mitochondrial small subunit ribosomal RNA gene (16S) (Table 2). For 16S, the following cycling conditions were used: (1) denaturation at 95 °C for 30 s; (2) denaturation at 95 °C for 30 s, annealing at 54 °C for 30 s, extension as 65 °C for 1 min (35 cycles); (3) final extension at 65 °C for 7 min. After a preliminary analysis of 16S data, at least one specimen from each major clade was chosen for amplification using our newly designed Amphinomida-specific 28S primers (Table 2). For amplification of a fragment of 28S (D1-D2 region), the primers 28S_Amph_F1 and 28S_Amph_R were used with the following cycling conditions were used: (1) denaturation at 95 °C for 30 s; (2) denaturation at 95 °C for 15 s, annealing at 64 °C for 20 s, extension as 72 °C for 45 s (35 cycles); (3) final extension at 72 °C for 7 min. For sequencing, the aforementioned 28S primers as well as the internal primer 28S_Amph_F2 were used to ensure adequate coverage of this relatively long amplicon (~954 bp). Agarose gel electrophoresis was used to check for PCR success. Most PCR reactions produced one clean band that could be directly purified using the Omega Bio-tek EZNA Cycle Pure Quick Kit. For PCR products with multiple bands, it was necessary to cut the correctly sized amplicon out of the gel and purify it using the Omega Bio-Tek MicroElute Gel Extraction Kit. The concentration of the purified PCR products was measured using a Nanodrop Lite. Notably, purified PCR products for 28S had high concentrations, with many over 50 ng/μL, and were diluted to 1 ng/μL after confirming the concentration with the Qubit ds DNA BR kit. Any PCR products measuring over 20 ng/μL on the Nanodrop Lite were diluted by half. Purified PCR products were sent to GeneWiz for Sanger sequencing. Sequencing was performed using the premix option with 10 μL of PCR product and 5 μL of 5 mM primer

for each reaction. DNA sequences were assembled into contigs, inspected, and manually edited for quality if needed using Sequencher version 5.4.6. All sequences were checked against the NCBI Nucleotide database using BLAST to determine closest match. Samples whose closest match was not a euphrosinid were excluded from the analysis (see below).

**Table 2.** Sequences of primers used for DNA amplification.

| Primer Name | Sequence 5′ to 3′ |
| --- | --- |
| 16S arL | CGCCTGTTTATCAAAAACAT |
| 16S brH | CCGGTCTGAACTCAGATCACGT |
| 28S_Amph_F1 | ACCCGCTGAAYTTAAGCATATCAC |
| 28S_Amph_F2 | ACAAGTACCGTGAGGGAAAGTTG |
| 28S_Amph_R | CTTGGTCCGTGTTTCAAGACG |

*2.4. Phylogenetic Analysis of Euphrosinidae (16S + 28S)*

A preliminary maximum likelihood analysis was performed in IQ-TREE 2 using the best-fitting model of sequence evolution and 1000 replicates of rapid bootstrapping for the 16S gene [18] (command used: iqtree2 -m MFP -B 1000 -s 16S.fas). The results of this phylogenetic analysis were used to determine which samples to sequence for the 28S gene and for species delimitation analyses (see below). For sequencing 28S, at least one specimen from each clade was chosen based on the quantity of available DNA.

All sequences were aligned with Muscle in MEGA 11 [19,20] and concatenated with FASconCAT-G [21]. IQ-TREE 2 was run on the resulting partitioned dataset as described above with the best-fitting model for each gene and unlinked branch lengths between the two partitions [18,22] (command used: iqtree2 -m MFP -B 1000 -Q FcC_supermatrix_partition.txt -s FcC_supermatrix.fas).

Species delimitation was performed on the 16S dataset using three approaches: (1) the Automatic Barcode Gap Discovery (ABGD) method with the 16S alignment as input, simple distance and default parameters on the ABGD web server (https://bioinfo.mnhn.fr/abi/public/abgd/abgdweb.html; accessed on 7 September 2022) [23], (2) Assemble Species by Automatic Partitioning (ASAP) with the 16S alignment as the input file, simple distance, and default parameters on the ASAP web server (https://bioinfo.mnhn.fr/abi/public/asap/asapweb.html; accessed on 7 September 2022) [24], and (3) multi-rate Poisson Tree Processes (mPTP) with the 16S contree file from the preliminary 16S analysis in IQ-Tree 2 as the input file and default parameters on the mPTP web server (https://mptp.h-its.org; accessed on 7 September 2022) [25] *Eurythoe_complanata* was used as the outgroup for all species delimitation analyses.

## 3. Results and Discussion

### 3.1. Diversity and Phylogeny of Euphrosinidae

We imaged and generated sequences for 59 specimens representing approximately eight species of North Atlantic Euphrosinidae from the IceAGE material as well as three Antarctic specimens representing two species, and *Euphrosine aurantiaca* from Southern California. We obtained additional sequences from GenBank when available. All specimens have a 16S sequence except for one specimen of *Euphrosine armadillo*, which was retained because its closest putative relatives were also sampled for 28S. Seventeen specimens also have a 28S sequence. Taken together, our final phylogenetic analysis included 82 euphrosinid sequences and is the most comprehensive DNA barcoding and phylogenetic study of Euphrosinidae to date (Figure 3).

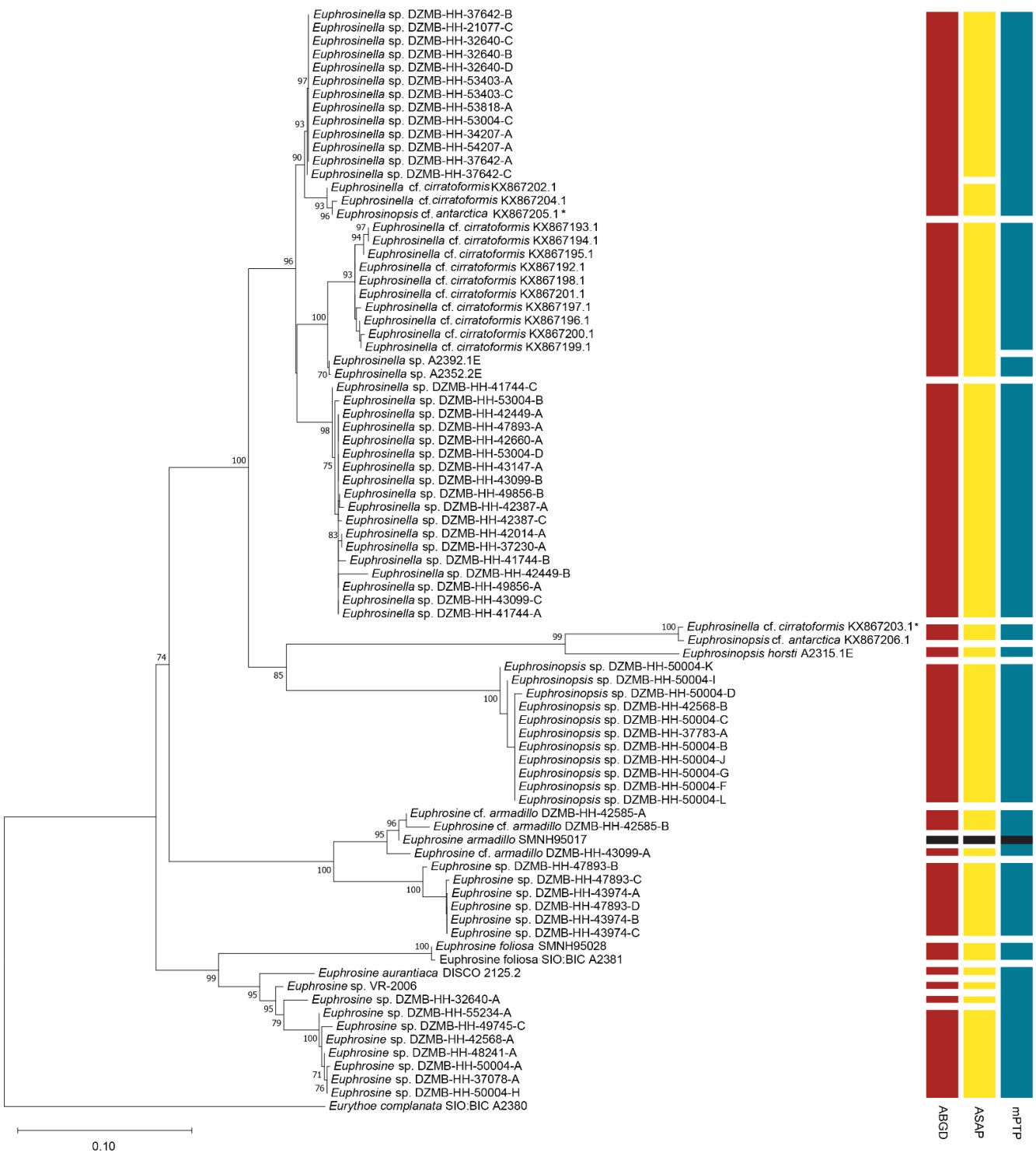

**Figure 3.** Phylogenetic analysis of Euphrosinidae based on 16S and 28S with results of three different species delimitation methods. Bootstrap support values below 70 are not shown. Asterisks indicate publicly available sequences where the original identification appears to be incorrect. Note that DZMB- numbers represent field collection codes and not museum catalaog numbers. Because of their small size, most of these specimens were destructively sampled for DNA extraction and thus museum catalog numbers are not available for these specimens.

Using ABGD species delimitation, we inferred a barcode gap distance of 3.4% and a total of fourteen Euphrosinidae species. Excluding the outgroup, the average p-distance between ABGD-inferred species ranged from 4.9% to 29.0% whereas average p-distance

within species ranged from 0% to 1.2% (see Supplementary Materials). Results of ASAP were nearly identical to those of ABGD except this method grouped three Antarctic *Euphrosinella* cf. *cirratoformis* specimens (KX867205.1 [presumably mislabeled as *Euphrosinopsis* cf. *antarctica*], KX867204.1, KX867202.1) in the same putative species as the clade containing 13 North Atlantic *Euphrosinella* specimens (e.g., DZMB-HH-37642-B). The third method tested, mPTP, inferred just 11 species of Euphrosinidae. This method grouped species similarly to ABGD except it grouped four *Euphrosine* lineages inferred to represent separate species by the other methods (*Euphrosine aurantiaca*, *Euphrosine* sp. VR-2006, *Euphrosine* sp. DZMB-HH-32640-A, and the clade containing *Euphrosine* sp. DZMB-HH-55234-A, etc.) as one species, it grouped all the specimens identified as *Euphrosine armadillo* or *Euphrosine* cf. *armadillo* as one species, and 3) it grouped the two newly sequenced Antarctic *Euphrosinella* specimens (A2352.2E and A2392.1E) as a separate species from the largest clade of *Euphrosinella* cf. *cirratoformis* (e.g., KX867193.1). Considering our morphological observations, the ABGD species delimitation method most accurately recapitulated morphospecies, although mPTP grouped specimens identified or tentatively identified as *Euphrosine armadillo*. Based on the ABGD species delimitation, ten North Atlantic species were sampled. For comparison, prior to this study, only two euphrosinids from the North Atlantic (*Euphrosine* sp. VR-2006 and *Euphrosine armadillo*) had publicly available DNA sequence data [26,27]. Notably, three of the sampled North Atlantic species were sampled from only one location, indicating that future sampling in the region may uncover additional species with geographically or bathymetrically limited distributions.

In our phylogenetic analysis of the combined 16S + 28S dataset, *Euphrosinella* was strongly supported to be monophyletic with a bootstrap support value of 96%. Currently, only two species of *Euphrosinella*, *E. paucibranchiata* and *E. cirratoformis*, are described [5]. *Euphrosinella paucibranchiata* was originally described from California and has only been seen in the Pacific Ocean [2]. *Euphrosinella cirratoformis* has been previously documented in the Southern Ocean and Southwest Pacific Ocean [2,3]. ABGD species delimitation identified three species of *Euphrosinella*, two of which are undescribed. Specimens conforming to *E. cirratoformis* were previously sequenced from the Southern Ocean [8] but specimens identified by that study as *E.* cf. *cirratoformis* represent two different species according to ABGD (three if the presumably misidentified specimen nested within *Euphrosinopsis* is considered). As *E. cirratoformis* has previously only been found in the Southern Ocean, and our ABGD and mPTP species delimitation analyses group North Atlantic specimens with one of the clades of Antarctic *E.* cf. *cirratoformis* (see top-most clade in Figure 3), our results raise the possibility of an extremely broad range for this species [2,8]. However, none of the Antarctic *E.* cf. *cirratoformis* sequences obtained from GenBank are identical to sequences from our North Atlantic animals and it may be that these are two distinct, but closely related species that were not correctly delimited by these methods as they were recovered as two separate species by ASAP species delimitation. Unfortunately, morphological evidence gathered from ethanol-fixed specimens was insufficient for us to confidently identify any of these specimens to the species level. Examination with the stereomicroscope and SEM did reveal that all North Atlantic specimens recovered in the *Euphrosinella* clade lack ringent chaetae, which is consistent with previous descriptions of the genus [2]. All *Euphrosinella* specimens investigated in this study also have two pairs of eyes as is consistent with described *Euphrosinella* (but also *Euphrosine*) [2].

*Euphrosinella* and *Euphrosinopsis* were recovered as sister taxa with a support value of 100%. *Euphrosinopsis* is supported as monophyletic with a modest bootstrap support value of 85%. The *Euphrosinopsis* clade includes sequences from two previously identified species and eleven new North Atlantic samples inferred to be the same species by all species delimitation methods. Based on branch lengths, *E. antarctica* and *E. horsti* appear to be quite genetically distinct from the North Atlantic species. Currently, *Euphrosinopsis* contains only three described species [5]. Based on our phylogenetic analysis we can say our North Atlantic species is not *E. horsti* or *E. antarctica*, which are both Antarctic. The only other *Euphrosinopsis* species, *E. crassiseta* lacks ringent chaeta, a feature that was clearly

visible in SEM examination of our North Atlantic samples. Therefore, our North Atlantic samples appear to represent a new species of *Euphrosinopsis*.

Surprisingly, *Euphrosine* was not recovered monophyletic, but rather as a paraphyletic grade, albeit with weak bootstrap support (bs = 74). Each of the two clades of *Euphrosine* has a sequence from at least one previously identified *Euphrosine* species that uncontroversially conforms to the diagnosis of the genus and our morphological examinations of the newly sequenced specimens were also consistent with the diagnosis of *Euphrosine*, confirming that both clades correspond to the genus *Euphrosine*. The clade containing *E. foliosa* and *E. aurantiaca* is monophyletic with a bootstrap support value of 99% and was recovered as sister to all other euphrosinids. The clade containing *E. armadillo* was recovered sister to *Euphrosinopsis* + *Euphrosinella*, but with weak support (bs = 74). Morphological descriptions of described *Euphrosine* species and the specimens sampled in this study were examined, but no morphological synapomorphies for either of the two clades of *Euphrosine* could be identified based on available data. The sequential branching of the two clades of *Euphrosine* as sister to all other sampled euphrosinids suggests that characteristics of the genus *Euphrosine*, such as the presence of two pairs of eyes, a prostomium with three appendages, and a caruncle attached to body wall, may be plesiomorphic for Euphrosinidae as a whole but more data are needed to test this phylogenetic hypothesis.

### 3.2. Analysis of Contaminant Sequences

Some specimens yielded high-quality sequences that, according to a BLAST [28] comparison against the NCBI Nucleotide database, were not from a euphrosinid and thus were excluded from the analysis. Two samples that yielded non-euphrosinid sequences had another group of annelid as the closest match. Of the two annelid contaminants, one (DZMB-HH-42697-A; from the North Atlantic) was from a species of Syllidae (ON228465) and the other (A2315.1E; from Antarctica), which was amplified and sequenced twice with the same results, was from a species of Cirratulidae (ON228459). Syllids can be found throughout the North Atlantic in diverse habitats [29]. Cirratulids are well-known from Antarctica and often live in burrows in muddy sediments and share habitats with euphrosinids [30,31]. Because amphinomids were the only annelids under study in the lab at the time this work was performed and given the obvious morphological differences among these different annelid clades, we are confident that we did not mistake a syllid or cirratulid for a euphrosinid. We speculate that this contamination may represent gut contents from these euphrosinids. Little is known about the euphrosinid diet, but some previous studies indicate they eat foraminiferans, sponges, bryozoans, and corals [32,33]. Our data suggest that other annelids may also be on the menu. An alternative hypothesis is that tissue fragments from other annelids collected during the same sampling event were caught in the chaetae of these euphrosinids and preferentially amplified during PCR. Future studies involving dissection or histological sectioning to examine diet would be of interest to improve understanding of euphrosinid feeding ecology.

### 4. Conclusions

The euphrosinid fauna of the North Atlantic is a diverse representation of the group with at least three of the four genera present in the region. As morphological differences between the genera *Euphrosinella* and *Euphrosine* are often small and easily overlooked [2], the establishment of a library of Euphrosinidae DNA barcodes could aid in identification of specimens collected in future environmental surveys, particularly when specimens are damaged. Examination of formalin-fixed (or even living) specimens from the region would likely enable formal description of new species of *Euphrosinella*, *Euphrosinopsis*, and potentially *Euphrosine*. Further, three species in our analysis are represented by individuals from only one location, indicating that future sampling in the region may uncover additional uncommon and/or geographically restricted species. Surprisingly, the genus *Euphrosine* was recovered as two distinct sequentially branching clades, thus rendering the group paraphyletic and suggesting that the characteristics of the genus *Euphrosine* may

be plesiomorphic for Euphrosinidae as a whole. However, given relatively weak support for this result, this hypothesis requires further testing with expanded taxon and molecular marker sampling.

**Supplementary Materials:** The following supporting information can be downloaded at: https://www.mdpi.com/article/10.3390/d14110996/s1, Table S1. Sampling locations, collection dates, and field collection numbers of specimens used in this study.

**Author Contributions:** Conceptualization, K.M.K.; formal analysis, R.A.B.; investigation, R.A.B. and K.N.W.; resources, K.M.K. and K.M.; funding acquisition, K.M.K.; data curation, R.A.B., K.N.W. and K.M.; writing—original draft preparation, R.A.B.; writing—review and editing, K.M.K., K.M. and R.A.B.; All authors have read and agreed to the published version of the manuscript.

**Funding:** This research was funded by National Science Foundation grant #1846174 to KMK. This publication was made possible with the help from the Alabama Water Institute at The University of Alabama.

**Institutional Review Board Statement:** Not applicable.

**Informed Consent Statement:** Not applicable.

**Data Availability Statement:** Sequence data for 16S and 28S including contaminant sequences were uploaded to the NCBI Nucleotide database (accession numbers ON148016-ON148085, ON228459 and ON228465). Photos and scanning electron micrographs of all specimens sequenced for this study were uploaded to FigShare (https://doi.org/10.6084/m9.figshare.21564063.v1) (accessed on 7 September 2022). Voucher specimens have been deposited at Senckenberg Museum Frankfurt and the Alabama Museum of Natural History. DNA extractions for all specimens sequenced (and others from which were unable to obtain sequence data using these methods) have been deposited into the Alabama Museum of Natural History (ALMNH:Inv:23681-23768). This collection is searchable via Arctos at http://arctos.database.museum/almnh_inv (accessed on 7 September 2022).

**Acknowledgments:** We thank Jerry Kudenov for assistance in specimen identification and helpful discussions about euphrosinid biology. We thank Leslie Harris (Los Angeles County Museum) for identifying and sharing the *Euphrosine aurantiaca* sample, which was collected through the DISCO project. We thank the crew and scientists of the IceAGE cruises aboard R/V *Meteor* and *Poseidon* and the crew and scientists, especially Ken Halanych, of the NBP 12-10 Cruise aboard the R/V *Nathaniel B. Palmer*. Finally, we thank three anonymous reviewers and the editors who provided constructive feedback that helped improve this manuscript.

**Conflicts of Interest:** The authors declare no conflict of interest.

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
