# Peer review of "Biodiversity and Phylogeny of North Atlantic Euphrosinidae (Annelida)"

_diversity, doi:10.3390/d14110996_

Round 1

Reviewer 1 Report

Please see attached documents.

Reviewer 2 Report

This is a fine manuscript, very well-written and concise. It can be published as it is presented to the journal.

The manuscript contains several interesting findings (paraphyly of Euphrosine, putative feeding on other polychaetes etc.). Most importantly the authors make a range of barcode sequences available, which will facilitate future taxonomic and population genetic work on Euphrosinidae. 

I only have very few trivial suggestions, which should not require much time to consider. 

  • not sure if the text citation for Figure 1 in line 31 is fitting. Might be better to move this to line 34 as the SEMs mostly show the chaetae.
  • re. calcareous chaetae -  Müller et al. 2021 (https://doi.org/10.1111/ivb.12353) is probably a better reference than Rouse & Pleijel 2001, as it includes EDS of euphorinsinid & amphinomid chaetae
  • line 140: GeneWiz is misspelled.
  • line 264: all three of the main genera - rephrase as three out of four genera
  • Author contributions: E.B.? I guess should be changed to R.B.

Reviewer 3 Report

This is a well written and important manuscript about the diversity of a little known annelid group.

I recommend this for publication following minor revision.

1. Although it is always nice to see SEM images, I do not see the use of Figure 1 as it is not referred to much in the text. I would suggest removing the image unless the authors either broaden the figure to include whole-worm image, as they only cite it once in one very general sentence on row 30-31, or discuss more about the differences in chaetae type between the three genera to make use of all those chaetae images. 

2. A follow-up on above is, if the authors choose to keep the figure, please use more informative specimen names in the figure text. This is also my suggestion for the phylogenetic tree, please label your separate species something unifying rather than just using the voucher numbers, to make it easier for the readers to follow and not having to browse back to find out which genus and species it refer to.

3. On row 57 the authors write ”Notably species of Euphrosinella….”, please expand a bit - all species of Euphrosinella or just one or two? 

4. In the Introduction the authors say that preservation in ethanol makes it impossible to properly identify species, and later on suggest that formalin-fixed samples are needed for that. However, the authors also state that DNA barcoding is useful to identify cryptic species, which makes the formalin-fixing a bit dubious - how to decide which species the formalin-fixed specimen belongs to without DNA? 

Furthermore, in most cases the authors have quite a lot of specimens for each DNA-species, studying them all might aid as the same appendages might not have fallen off every specimen? Another solution is to actually live-sort the samples onboard (yes, time consuming, but rewarding) and fix the specimens separately in 80% ethanol, which keeps the morphology better while still being good for molecular work.

5. In the file, I did not get Table 2, but in the material and methods I would suggest a line or two stating more in-depth where in Antarctic waters the additional specimens were collected.

6. As not the whole of 28S was sequenced, it would be very useful to know which parts were amplified. 28S is divided into 10 regions, D1-D10, and for other users doing only parts of 28S it’s always good to know which published sequences would match theirs without having to first download all sequences and try to align them.

7. I do not understand the sentence on lines 183-185, why would the grouping of one of the authors Euphrosinopsis with another Euphrosinopsis species not be consistent with the authors identifications?

8. On line 226 the authors state ”Surprisingly, Euphrosine was not recovered monophyletic…”, but when looking at the tree the support values for those basal clades are way too low to state anything about relationships at that level. I would suggest the authors remove the statement or else add in that the support values are too low to give any certainty. Generally, support values below 80% are not credible in Maximum Likelihood analyses.

9. The sentence about syllids on line 248 seems a bit odd, does the authors mean that because the syllids move fast they can’t be eaten by slower animals? Maybe expand a bit or exclude this sentence.
